# Relating GPI-Anchored Ly6 Proteins uPAR and CD59 to Viral Infection

**DOI:** 10.3390/v11111060

**Published:** 2019-11-14

**Authors:** Jingyou Yu, Vaibhav Murthy, Shan-Lu Liu

**Affiliations:** 1Center for Retrovirus Research, The Ohio State University, Columbus, OH 43210, USA; yu.2123@osu.edu (J.Y.); murthy.39@buckeyemail.osu.edu (V.M.); 2Department of Veterinary Biosciences, The Ohio State University, Columbus, OH 43210, USA; 3Department of Microbial Infection and Immunity, The Ohio State University, Columbus, OH 43210, USA; 4Viruses and Emerging Pathogens Program, Infectious Diseases Institute, The Ohio State University, Columbus, OH 43210, USA

**Keywords:** Ly6/uPAR, viruses, Ly6E, uPAR/CD87, CD59

## Abstract

The Ly6 (lymphocyte antigen-6)/uPAR (urokinase-type plasminogen activator receptor) superfamily protein is a group of molecules that share limited sequence homology but conserved three-fingered structures. Despite diverse cellular functions, such as in regulating host immunity, cell adhesion, and migration, the physiological roles of these factors in vivo remain poorly characterized. Notably, increasing research has focused on the interplays between Ly6/uPAR proteins and viral pathogens, the results of which have provided new insight into viral entry and virus–host interactions. While LY6E (lymphocyte antigen 6 family member E), one key member of the Ly6E/uPAR-family proteins, has been extensively studied, other members have not been well characterized. Here, we summarize current knowledge of Ly6/uPAR proteins related to viral infection, with a focus on uPAR and CD59. Our goal is to provide an up-to-date view of the Ly6/uPAR-family proteins and associated virus–host interaction and viral pathogenesis.

## 1. Introduction: Biosynthesis, Structure, and Functions

The Ly6 (lymphocyte antigen-6)/uPAR (urokinase-type plasminogen activator receptor) superfamily proteins were initially identified as T cell antigens in activated murine T lymphocytes by alloantisera staining [1,2]. The first molecular cloning of Ly6 cDNA was carried out in 1986, revealing a group of genes in the *Ly6* gene in murine chromosomes 15 [3]. Since then, multiple genes in the Ly6 family have been isolated, including murine LY6A [4], LY6C [4], LY6E [5], LY6I [6], among others (Table 1). Human orthologs were isolated shortly after and most of these genes were mapped to human chromosome 8 [7] (Table 1). To date, *Ly6/uPAR* genes have been discovered in insects [8], fish [9], amphibians [10], reptiles [11], birds [12], and mammals [13] (Table 1). The general knowledge of the Ly6/uPAR family, including their genomic organization, tissue distribution, and evolution, has been elegantly reviewed elsewhere (refer to [7,14,15,16]).

There are more than 30 genes that have been classified into the LY6/uPAR superfamily [7]. The proteins encoded by the *LY6/uPAR* genes share at least one conserved functional motif, known as the LY6/uPAR (LU) domain (Figure 1a,b). The LU domain adopts a “three-fingered” folding topology characterized by 4–5 consensus disulfide bonds and an invariant carboxyl-terminal (C-terminal) asparagine. Interestingly, the length as well as the amino acid sequences aligned at the fingertips are divergent, which renders the three-finger structure flexible for a broad range of intermolecular interactions [7]. In addition to the LU domain, Ly6/uPAR family proteins also harbor a conserved “LXCXXC” motif at the amino-terminus (N-terminus) and a “CCXXXXCN” motif at the carboxyl-terminus (C-terminus) [7] (Figure 1). The “LXCXXC” motif is thought to be the binding site for transition metal ions [18] while the function of the “CCXXXXCN” motif is less well defined.

Similar to many membrane-associated proteins, the LY6/uPAR family proteins are initially synthesized in the form of a precursor, which contains an N-terminal signal peptide (SP), an LU domain(s), and a C-terminal glycosylphosphatidylinositol (GPI) moiety anchor in most cases (Figure 1). The N-terminal SP is rapidly removed by peptidase in the endoplasmic reticulum (ER) upon translocation, while the C-terminus GPI is appended via transamidase in the ER through the conserved asparagine of the nascent protein [19]. The glycolipid GPI-anchoring requires a specific signal, which can either be a consensus motif and/or the length of amino acids following an asparagine residue [20,21]. Because the GPI moiety-carrying hydrophobic modification has a high affinity to lipid rafts, GPI-anchored proteins are often associated with lipid raft-enriched microdomains in the membrane [20]. Notably, certain LY6/uPAR proteins, such as SLURP1 (secreted Ly-6/uPAR-related protein 1) [22] and SLURP2 (secreted Ly-6/uPAR-related protein 2) [23], do not have a GPI anchor because of the lack of a GPI addition motif, and as a result, these proteins are secreted following the canonical protein secretion pathway. Noticeably, some LY6/uPAR-family proteins can form dimers or multimers via covalent or non-covalent binding [24,25,26], which collectively execute biological functions.

The function of Ly6/uPAR has been historically linked to immunoregulation, including T lymphocyte development [27], differentiation [28], activation [29], proliferation [30], and migration [16], most of which were studied in mice. Interestingly, clinical investigations of Ly6/uPAR in humans, however, have revealed some distinct pathological functions. For example, increased LY6E expression is associated with solid tumorigenesis, angiogenesis [31], systemic lupus erythematosus [32], and other abnormalities [33,34]. In contrast, regulation of Ly6/uPAR proteins by virus infection, and vice versa, is not well understood, and there is an emerging interest in understanding how these families of proteins influence the process of viral infection.

## 2. Regulation of Ly6/uPAR Expression by Cytokines and Viral Infections

Expression of many Ly6/uPAR-family proteins is induced by immune-regulated cytokines, including those triggered by viral infections (Table 2). For example, murine LY6A, LY6C, and LY6E are up-regulated in T lymphocytes by recombinant human interferon (IFN) α, β, and γ [35,36,37]. LY6C is enhanced by cytokines interleukin 27 (IL-27) and augmented by T-cell receptor (TCR) stimulation [38]. Human LY6E is characterized as a typical IFN-inducible protein or defined as an IFN-stimulated gene (ISG) [39,40,41]. In monocytes and monocyte-derived THP-1 cell line, LY6E is up-regulated by treatment of cells with 100 U/mL IFNα as early as 6 h [42,43]. Interestingly, another LY6 family member, SLURP-2 is enhanced by IL-22 treatment, and this effect can be completely abolished by IFN-γ treatment [44]. The inducibility of LY6 family proteins by IFN is believed to be related to the IFN sensitive cis-acting elements within their promoter regions; however, a mechanistic study has found that the IFNγ-activating site (GAS), instead of canonical IFNα-stimulated response element (ISRE) in the LY6 gene promoter region, is responsible for induction by IFN [45].

Given that the LY6/uPAR-family proteins can be induced by type I IFN and cellular inflammation systems, it is not surprising that viral infection, which itself triggers the type I IFN production and inflammatory response, can induce or activate the *LY6/uPAR* gene expression. Indeed, LY6E has been widely associated with inflammation-related abnormalities, including systemic lupus erythematosus (SLE) [32,46], solid cancer [47,48], and viral infections [49,50].

In this review, we focus on two members of the Ly/uPAR-family proteins, uPAR and CD59, in the context of their effects on viral infection. In an accompanying review of this special issue, the role of LY6E in virus-host interaction, particular viral entry, is discussed [51].

## 3. uPAR and Viral Infection

uPAR, also known as CD87, is a heavily glycosylated, GPI-anchored cell-surface receptor [62,63]. It is predominantly expressed in immune cells, including neutrophils, monocytes/macrophages, and activated T cells [64]. uPAR harbors three consecutive LU repeats, namely D1, D2, and D3, respectively. The N-terminal D1 is responsible for the binding to urokinase-type plasminogen activator (uPA), the ligand of uPAR [65], and the linker peptides connect different repeats and contribute to chemotaxis [66].

Apart from the full length, several other forms of uPAR have been identified in conditioned medium from various cell lines [67], as well as in body fluids of cancer patients [68,69]. First, the intact uPAR (D1D2D3) is tethered to the cell surface by a GPI anchor attached to D3. Second, proteolytic cleavage in the linker region between D1 and D2 results in the release of the D1 fragment, leaving behind the D2D3 fragment on the cell surface. Third, soluble forms, which lack the GPI anchor but harbor either soluble uPAR (suPAR) or D2D3, can be generated by phospholipase C cleavage of the GPI anchor. In physiological settings, uPAR functions mainly through binding to its cognate ligand Urokinase-type plasminogen activator (uPA). uPA is a specific protease, which converts the plasminogen into its active form, plasmin—a broad-spectrum serine protease involved in the digestion of basement membranes and various protein substrates in the extracellular matrix [70,71]. The binding of uPA, which can be endogenously produced or released from surrounding cells, to uPAR concentrates the plasmin proteolytic activity on the relevant cell surface [72]. Therefore, the uPA/uPAR system plays a crucial role in cell migration and extravasation. Additionally, uPAR has been widely associated with vascular homeostasis, inflammation, tissue repair, cell adhesion and migration, signal transduction, tumorigenesis and metastasis, the scope of which has been elegantly reviewed elsewhere [73].

Investigation into the interplay between uPAR and viral infection can be traced back to the early 1990s, with a strong bias toward HIV studies (Table 3). It was reported that HIV-1 infection led to an enhanced cell surface expression of uPAR in monocytes and T lymphocytes in vitro and in vivo [74,75]. HIV-1 infection in tonsil histocultures significantly increased the suPAR expression in the culture medium [76]. Subsequent studies showed that uPAR mRNA was transcriptionally elevated in the context of HIV-1 infection [75]. However, how HIV-1 modulates the uPAR mRNA has remained unclear. One possibility is that HIV-1 infection may indirectly enhance uPAR expression through immune activation. It has been reported that uPAR expression is intimately regulated by some inflammation-inducing ligands, such as microbial components [77,78], mitogens [74], and pro-inflammatory cytokines [79,80], and that HIV-1 infection is associated with sustained chronic immune activation and inflammation [81,82]. Interestingly, increased uPAR has also been observed in pathological conditions, such as diabetes [83], cardiovascular disorders [84], cancers [85], and live diseases [86]. However, decreased uPAR expression has also been reported in granulocytes of HIV infected patients [87], suggesting that HIV-1 infection may modulate the immune system in a cell type-specific manner.

Elevated expression of uPAR has long been associated with HIV-1 disease progression and AIDS-related deaths [88], with suPAR level in serum of untreated HIV-1 patients being significantly higher than those of healthy cohorts [88]. In addition, the suPAR level in cerebrospinal fluid (CSF) is positively correlated with the progression of HIV-1 induced central nervous system (CNS) complications [89,90]. HIV-1 positive individuals also have enhanced cell-associated uPAR in lymphoid organs, particularly in follicular dendritic cells, macrophages, and endothelial cells [76]. Higher levels of suPAR, and to a lesser extent uPA in the cerebrospinal fluid (CSF) of HIV-positive patients, were also observed compared with HIV-negative controls [90,91]. These correlative studies collectively suggest a functional interplay between HIV-1 infection and the uPA/uPAR system, strongly implicating a positive role of uPAR in HIV-1 infection.

Some in vitro and ex vivo studies have provided mechanistic insights into how uPAR might enhance HIV infection. First, uPAR synergistically functions with uPA to promote HIV infection. As a serine protease, uPA specifically cleaves peptides that harbor the consensus cleavage motif, i.e., CPGRV, which is present and conserved in the HIV envelope protein (Env) gp120 variable loop 3 region (V3) [92]. Consequently, incorporation of uPA into HIV-1 virion can lead to aberrant enzymatic processing of Env. Somewhat surprisingly, however, this cleavage facilitates CCR5-tropic HIV-1 infection of human macrophages, probably by increasing viral fusion [92]. Clinical studies revealed that HIV-1 pathogenesis is associated with uPAR but not uPA expression [88,89], which argues against a direct role of uPA in HIV-1 infection. One possible explanation is that the plasma membrane-residing uPAR, which helps concentrate uPA on the cell surface [93], might render greater spatial proximity of the latter to the viral budding sites and thus increases the efficiency of uPA incorporation into virions and subsequent cleavage.

Second, uPAR can facilitate HIV-1 cell-to-cell spreading. It has been reported that uPA/uPAR interaction triggers signaling cascades in macrophages [93], leading to RhoA and PKCξ-dependent actin rearrangement [94] and subsequent intracellular enrichment of HIV-1 in a specialized structure called virus-containing compartments (VCCs) [94,95]. VCCs are regarded as invagination of the macrophage plasma membrane, which is usually connected to the extracellular space via tubular channels, and serve as the primary assembly and budding sites of HIV in macrophages [96,97]. Recently, VCC was shown to be an immune-privileged site for anti-HIV therapy [98] and antibody neutralization treatments [99]. Therefore, VCCs induced by uPA/uPAR may function as an immune privileged niche that protects HIV from a hostile environment both within and outside of the cells [97]. Additionally, VCCs can serve as HIV-1 reservoirs in macrophages and contribute to HIV-1 cell-to-cell transmission by translocating the inside virion cargoes into T cells through the virological synapse, a transient intercellular adhesive structure formed between infected and uninfected HIV target cells [100,101]. Given that the HIV cell-to-cell transmission is much more efficient than the cell-free infection [102], it is conceivable that uPA/uPAR signaling likely leverages the route of HIV to cell-to-cell transmission; however, details of this process warrant further investigation.

Third, uPA/uPAR can promote HIV transmission by enhancing macrophage adhesion [103,104], chemotaxis [66,73], and motility [105]. It is well-known that HIV spread, in particular in vivo transmission, involves the physiological contact of macrophages with other cell types; therefore, increased adhesion and migration would allow more efficient cell–cell contact formation therefore benefit virus spread. In this sense, a friendly microenvironment, such as that created by uPA/UPAR for HIV-1 in vivo transmission, would exacerbate HIV pathogenesis.

In addition to HIV-1, increased plasma levels of uPA and uPAR have also been associated with acute and chronic hepatitis B virus infections [106]. Interestingly, in vivo studies conducted in mice showed a minor role of uPA/uPAR in limiting the virus replication and in orchestrating the innate immune response to infection by the human respiratory syncytial virus (HRSV) and influenza virus [107]. While more investigations are needed to elucidate the role of uPA and uPAR in viral infection and pathogenesis, it is possible that the effect of uPA/uPAR in the context of viral infection is virus-specific and cell type-dependent.

## 4. CD59 and Viral Infection

CD59 (or protectin) is a non-interferon inducible protein initially identified as an inhibitor of complement-mediated lysis [135,136,137]. Preliminary sequence alignment showed limited homology with other Ly6 members; however, subsequent structural studies using PI-PLC cleavage indicated a cell surface linkage via GPI-anchor, as well as a distant evolutionary relationship with other members of the Ly6 family [135]. The most widely understood function of CD59 is its involvement in the disruption of the membrane attack complex (MAC) during complement-mediated lysis [138]. Specifically, CD59 acts in the final stages of MAC assembly by inhibiting C9 input to EC5b-8 and subsequently incorporating itself into the complex [138,139], presumably because of its similar binding pocket to the C8α-chain [140]. Further, this disruption is specific to complement as CD59 is not seen to disrupt perforin-mediated lysis [141]. Clinically, deficiency of CD59 is responsible for the development of paroxysmal nocturnal hemoglobinuria (PNH), as noted by increased susceptibility of erythrocytes to complement-mediated lysis [142,143]. Interestingly, cells from patients with PNH can re-acquire resistance to hemolysis when incubated with CD59 [144,145]. The involvement of CD59 in complement lysis and subsequently its dysregulation in PNH has been extensively characterized and reviewed by [146].

CD59 is expressed on a wide variety of cell types [147], and in accordance, has numerous non-complement-dependent roles. CD59 is accumulated in tumor cells [148], and there has been directly targeted and down-modulated by using monoclonal antibodies (mAbs), shRNA and other small molecules for therapeutics [149,150]. CD59 is also implicated in T-cell signaling [149,151] and regulates cell growth and apoptosis. For example, CD59 can coordinately interact with CD2 during T-cell activation and adhesion [152,153]. In CD3+ Jurkat cells, activation of ZAP-70 and p65lck results in activation of the T-cell receptor and downstream signaling to produce interleukin 2 (IL-2) [154]. This interaction is mediated via a mobile Lck fraction [155] and an adaptor protein linker for activation of T-cells (LAT) that is recruited to lipid rafts [147,151,152,153,154,155,156,157], which is the characteristic localization region for members of Ly6/uPAR proteins.

There is a growing body of literature showing the involvement of CD59 in viral infection and pathogenesis. CD59 has been shown to interact with numerous viruses, either directly or indirectly, through complement-dependent and independent mechanisms. Generally, these interactions can be classified into three categories: (1) Incorporation of CD59 into viral envelopes to evade complement virolysis; (2) modulation of CD59 on the cell surface to escape immune sensing; and (3) expression of virus-encoded CD59-mimic proteins.

By far the most characterized virus-CD59 interactions are with HIV-1/2 and related SIVs. Several reports have shown a decreased surface expression of CD59 on peripheral blood T-lymphocytes [122], erythrocytes [158], neuronal and astroglial cells [159] in HIV-1 patients. In the latter study, cells treated with recombinant gp41 showed a decreased CD59 level. While no direct evidence for increased complement-mediated damage was observed, the likely outcome of this down-modulation of CD59 is increased complement-mediated lysis. Indeed, when these cells were treated with phorbol dibutyrate (PdBu; an activator of protein kinase C-PKC), CD59 levels were decreased following treatment with recombinant gp41, suggesting a PKC-dependent signaling role. Similar results have been observed when pro-inflammatory cytokines (IL-1β, IFN-γ) and LPS are used in neuronal cell lines. Interestingly, subsequent experiments to identify complement activation in H9 cells by HIV-1 and HIV-2 isolates found no difference in CD59 levels [160]. However, when HIV particles were analyzed using virus capture assay, complement-inhibiting protein decay-accelerating factor (DAF) was found to be incorporated into viral particles, with CD59 also incorporated but to a lesser extent, which would lead to protection from complement-mediated lysis [118,120]. Similar complement-controlling proteins are seen in SIV particles, which likely helps infected cells escape from complement-mediated killing [119]. The incorporation of complement control proteins in HIV particles is not restricted to CD59, as both DAF (also known as CD55) and CD45 are incorporated into viral particles [121]. It should be noted that although early reports suggested that HIV-1 is insensitive to complement lysis, whether or not the incorporation of CD59 and other complement controlling proteins are sufficient to allow for this escape remains to be determined.

Incorporation of cellular proteins into virions and subsequent immune evasion has been reported for numerous host proteins. For example, CD59 is incorporated into viral particles of vaccinia virus [132], human T-cell leukemia/lymphoma virus (HTLV-1), human cytomegalovirus (HCMV) [110,161], infectious bronchitis virus (IBV) [130], hepatitis C virus [124], Newcastle disease virus [162], as well as parainfluenza virus 5 [134] among others (Table 3). In IBV-infected cells, the cell surface level of CD59 is down-regulated because of incorporation into viral particles. IBV production from CD59 knock-down cells was significantly reduced; along the same lines, when the GPI-anchor of CD59 is cleaved by using PI-PLC, the titer of IBV was also significantly reduced [130]. HCV particles purified from cell culture showed CD59 incorporation in the viral membranes [124], which is co-localized with HCV proteins during the assembly process [125]. Similar GPI-dependence has been observed in production of HIV-1 particles, where GPI-anchor deficiency rendered viruses more susceptible to complement-mediated lysis [117].

Intriguingly, herpesvirus saimiri (HVS) encodes a gene ORF15 with 64% nucleotide homology to CD59 [112]. Structural characterization showed that the HVS CD59 protein shares the single N-linked glycosylation with human CD59 and the highly conserved cysteine residues [41]. The N-linked glycosylation site is predicted to be responsible for the cross-species specific complement-resistance activity [113]. For example, human CD59 and saimiri CD59 are not cross-protective against rat complement, and HVS CD59 provides no protection against human or rat serum. This is likely attributed to the differentially located glycosylation site of HVSCD59 as compared to the conserved primate CD59 glycosylation sites [113]. While further studies are required to determine the exact mechanism of protection, or lack thereof, the viral mimicry of complement control proteins has been observed in numerous other viruses [163].

Antagonism of host-cellular proteins is a hallmark for productive infection in viruses. Downregulation of CD59 from the surface of virus-infected cells is a common cellular mechanism hijacked by viral particles to promote complement-mediated lysis either for viral egress or chronic disease manifestation. For instance, CD59 is significantly reduced in monocytes from DENV-infected patients [127]. Similar results have been seen in patients with chronic HBV infection. Using HBV transgenic mice expressing HBV genome, Qu et al. showed decreased levels of CD59 at both mRNA and protein levels [61]. Similar observations were seen in hepatocytes infected with HBV and this downregulation was specific to CD59, as neither CD55 nor crry (another complement controlling protein) were significantly changed upon infection. It is now known that the HBV core protein, which is responsible for this downregulation, sensitizes hepatocytes, thus resulting in lysis and subsequent liver damage [109]. HepG2 cells transfected with HBV core protein selectively form complexes with CD59 and they are recapitulated in liver samples from HBV patients. Interestingly, these complexes are seen to translocate to the nucleus, resulting in decreased surface expression and an overall increase in susceptibility to MAC [109].

The modulation of CD59 levels on the cell surface has also been employed by other viruses. For instance, varicella-zoster virus upregulates CD59 in thymus/liver and dorsal root ganglia (DRG) xenografts in-vivo. Interestingly, this upregulation is tissue or cell-specific, as similar infections in skin xenografts showed only modest upregulation [111]. The exact mechanism of this upregulation remains unclear. CD59 upregulation is thought to be a consequence of upstream NF-kB activation; however, VZV is known to inhibit NF-kB signaling in skin implants. Interestingly, antibodies against CD59 block infection to a range of echoviruses in RD cells. Noteworthy is that echovirus 7 uses DAF as a receptor for entry and that blocking CD59 does not affect the virus binding to DAF or cell-to-cell spread. It is hypothesized that CD59 acts at an early stage of virus entry but not during the attachment.

CD59 is involved in diverse and essential cellular functions, especially in its protection of “self” from complement-mediated lysis. Thus, it is not surprising that viruses have evolved strategies to exploit this function, either through incorporation of it into viral envelopes or by mimicking the functional capacity of CD59. CD59 also appears to be an attractive target for therapeutic strategies to prevent or control viral pathogenesis. Indeed, recent therapeutic efforts have focused on inhibiting CD59 by using hCD59 inhibitors [164], or antibodies combined with anti-HIV Env antibody or serum from HIV-1 infected individuals [165]. However, the challenge is that CD59 molecule is widely distributed on the cell surface and performs some important cellular functions. Therefore, strategies to specifically target CD59 on the virion membrane should be pursued.

## 5. Concluding Remarks and Future Perspectives

Some general conclusions can be drawn from the research performed on uPAR and CD59 proteins related to viral infections. First, Ly6/uPAR proteins can function through direct and indirect mechanisms. Second, Ly6/uPAR molecules influence viral infection in a cell context-dependent and virus-specific fashion. Last, the function of Ly6/uPAR is related to the GPI-anchored topology of these proteins, as well as their lipid-raft localizations.

Notably, most Ly6/uPAR-family members remain uncharacterized in the context of viral infection. This is especially important, given that some of these proteins may have a redundant, cooperative, or synergistic effect in viral infection. For example, knockdown of one Ly6/uPAR member may affect—either inhibit or enhance—the functions of others during viral replication. While GPI anchor is known to affect the protein location and interactions with others, how exactly the Ly6/uPAR proteins with a GPI anchor regulate cell signaling, migration, and other physiological processes involved in viral infection remain largely unknown. Hence, it would be interesting and informative to use global and comparative approaches to examine the effects of LY6/uPAR family members in the context of innate and adaptive immune response to viral infection. Finally, most of the published studies have been derived from in vitro or mouse experiments, roles of Ly6/uPAR in human viral infections need to be determined, including under pathological conditions. Additionally, genetic mapping and single nucleotide polymorphism (SNP) studies of human Ly6/uPAR genes will provide insights into the role of Ly6/uPAR-family members in viral infection and virus-host co-evolution.

## Figures and Tables

**Figure 1 viruses-11-01060-f001:**
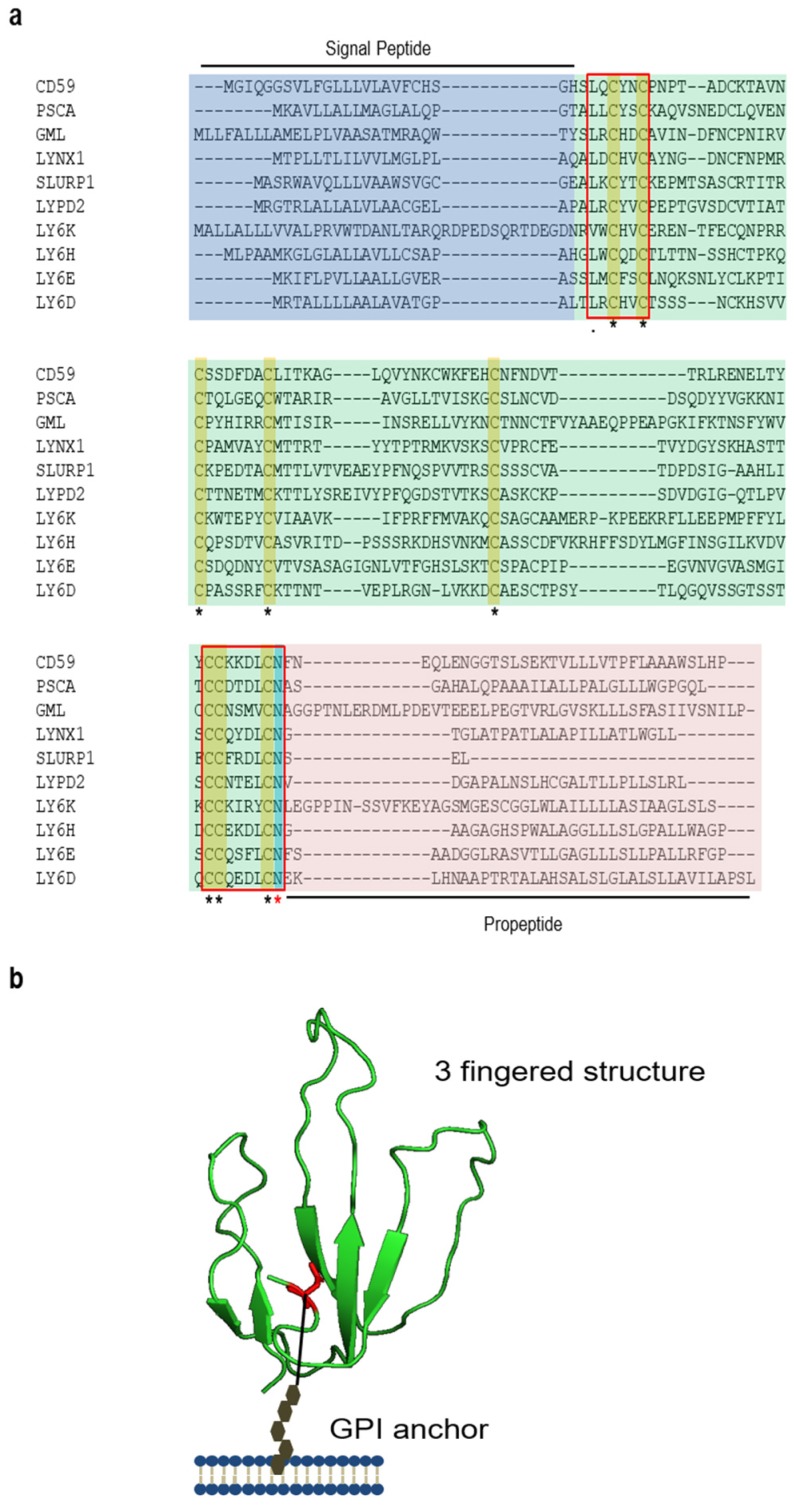
Sequence alignment and domain structures of LY6/uPAR family proteins. (**a**) Sequence alignment of major LY6/uPAR-family protein members. The shaded light blue box shows the signal peptide predicted by online software SignalP-5.0 (http://www.cbs.dtu.dk/services/SignalP/); shaded light green box shows the LU domain; and shaded light red indicates pro-peptides (GPI anchors), which are removed in mature peptides. Yellow color highlights eight conserved cysteine residues, while the cyan color shows the asparagine residue that can be glycosylated and linked to a GPI anchor. Red squares show two conserved motifs: the amino terminal “L/VXCXXC” and the carboxyl terminal “CCXXXCN”. (**b**) Domain structure of LY6E. Human LY6E was homology-modeled based on the submitted structure of SLURP-2 (PDB ID: 2MUO). Four disulfide bonds are shown in yellow while the GPI anchor is shown in black.

**Table 1 viruses-11-01060-t001:** Features of major Ly6/uPAR proteins.

Protein	Full Name	Virus Interaction	Tissue or Cell Expression	Species (Chromosome #)	Types of Protein	Other Alias
LY6A	Ly6 complex, locus A	↑ Mouse adenovirus type 1 (MAV-1); ↑ Adeno-associated virus (AAV) serotype 9 (AAV-9)	Hematopoietic stem cells, B cell, T cell, DCs	Mouse (15)	GPI-anchored	TAP; Sca-1; Ly-6A.2; Ly-6A/E; Ly-6E.1
LY6B	Ly6 complex locus B	Unknown	Neutrophils, inflammatory monocytes, and some activated macrophages	Mouse (15)	GPI-anchored	7/4; GM-2.2
LY6C1	Ly6 complex locus C1	Unknown	Inflammatory monocytes, some NK cells, and plasmacytoid dendritic cells	Mouse (15)	GPI-anchored	LY6C
LY6C2	Ly6 complex locus C2	Unknown	Leukemia cells and on macrophages infiltrating rejected allografts	Mouse (15)	GPI-anchored	
LY6D	Ly6 complex locus D	↑ HIV-1 [17]	B cells, immature thymocytes, and plasmacytoid dendritic cells	Human (8) Mouse (15)	GPI-anchored	Thb; Ly61
LY6E	Ly6 complex locus E	↑ Flavivirus: YFV, ZIKV, DENV, WNV ↑ Retrovirus: HIV or ↓ Rhabdo virus: VSV ↑ Orthomyxovirus: IAV	Most intrathymic precursor cells of the lymphoid lineage	Human (8) Mouse (15) Birds	GPI-anchored	RIG-E; Sca-2; TSA-1
LY6F	Ly6 complex locus F	Unknown	Nonlymphoid tissues	Mouse (15)	GPI-anchored	
LY6G	Ly6 complex locus G	Unknown	Mature granulocytes	Mouse (15)	GPI-anchored	Gr-1
LY6H	Ly6 complex locus H	Unknown	Brain	Human (8) Mouse (15)	GPI-anchored	NMLY6
LY6I	Ly6 complex locus I	Unknown	Spleen, thymus, kidney, and lung; bone marrow cells, monocytes, macrophages, granulocytes, and myeloid precursors	Mouse (15)	GPI-anchored	Ly6M
LY6K	Ly6 complex locus K	Unknown	Testis and keratinocytes	Human (8) Mouse (15)	GPI-anchored	
LYPD2	Ly6/Plaur domain-containing 2	Unknown	Esophagus, skin, and stomach	Human (8) Mouse (15)	GPI-anchored	VLL; Lypdc2
SLURP1	Secreted Ly6/Plaur domain-containing 1	Unknown	Restricted in esophagus	Human (8) Mouse (15)	Secreted	ARS
LYNX1	Ly6/neurotoxin	Unknown	Unknown	Human (8)	Secreted	SLURP2
GML	GPI-anchored molecule-like protein	↑ HIV	Adrenal gland	Human (8) Mouse (15)	GPI-anchored	HemT-3, LY6DL
PSCA	Prostate Sca	↑ YFV	Prostate	Human (8) Mouse (15)	GPI-anchored	
GP1HBP1	GPI-anchored HDL-binding protein 1	Unknown	Heart, lung, liver	Human (8) Mouse (15)	GPI-anchored	
uPAR	Urokinase plasminogen activator surface receptor	↑ HIV-1	Monocytes, dendritic cells, activated T and NK cells, endothelial cells, keratinocytes, and fibroblasts	Human (19) Mouse (7) Others	GPI-anchored	CD87, PLAUR
CD59	CD59 molecule	↑ HIV-1 ↑ HCV ↑ Cytomegalovirus ↑ infectious bronchitis virus (IBV)	Ubiquitously expressed; high in erythrocyte	Human (11) Mouse (2) Birds Amphibians Bony fishes	GPI-anchored	16.3A5, 1F5, EJ16, MAC-IP

Modification from reference [7,14,15,16]. ↑ and ↓ denote up- and down-regulation of viral infection, respectively.

**Table 2 viruses-11-01060-t002:** Regulation of Ly6/uPAR expression by cytokines and viral infections ^1^.

Protein Name	Viral Infection	Cytokine
LY6A	↑ JEV, WNV, and Reovirus [52]	↑ Recombinant human IFN α, β, and γ [35,36,37]
LY6C	WNV infection associated with lower LY6C expression [53]	↑ Recombinant human IFN α, β, and γ [35,36,37] IL-27 [38]
LY6E	↑ HIV-1infection [54]; ↑ SIV [55] ↑ JEV, WNV, and Reovirus [52]	↑ Recombinant human IFN α, β, and γ [35,36,37,42,43]; ↑ Retinoic acid [41]
LYNX1	Unknown	↑ IL-22 [44]
uPAR	↑ HIV-1gp120 in B cells [56]	↑TNF-α [57]; ↑ IL-1β [58]; ↑ Nerve growth factor [59]
CD59	↓ EBV [60] ↓ HBV [61]	Unknown

^1^ ↑ and ↓ denote up- and down-regulation of Ly6/uPAR proteins by virus infection or cytokines, respectively. JEV: Japanese encephalitis virus; WNV: West Nile virus; HIV-1: human immunodeficiency virus type 1; SIV: Simian immunodeficiency virus; EBV: Epstein-Barr virus; HBV: hepatitis B virus.

**Table 3 viruses-11-01060-t003:** Effects of LY6E and uPAR on viral infection.

Protein Name	Virus Name	Family of Virus	Effect on Infection	Mechanism of Action	Experimental System	Reference
**uPAR**	HIV-1	Lentivirus	Enhanced	1. Facilitate HIV-1 enzymatic processing of Env; 2. Promote HIV-1 cell-to-cell transmission; 3. Enhance macrophage adhesion.	Macrophages	[92,94,95,103,104]
	Human respiratory syncytial virus	Orthopneumovirus	Resistant	Unknown	C57BL/6 mice	[107]
	Influenza A virus (IAV)	Orthomyxovirus	Resistant	Unknown	C57BL/6 mice	[107]
**CD59**	HBV	Hepadnavirus	Enhanced	1.Promotes CDC to cause persistent liver inflammation; 2. Prevents CDC in hepatoma and hepatic cells that express HBV-X protein.	HBV BALB/c mice, BEL7402, HL7702, HepG2 cells	[61,108,109]
	Human cytomegalovirus (HCMV)	Herpesvirus	Enhanced	Incorporated into viral particles and confers CDC resistance.	Human Foreskin Fibroblasts (HFF)	[110]
	Varicella-zoster Virus (VZV)	Herpesvirus	Enhanced	Upregulated upon VZV infection to protect against CDC.	Human T-cells, xenograft SCID-hu mice, satellite glial cells,	[111]
	EBV	Herpesvirus	Resistant	Decreased CD59 expression to allow for CD8+T-cell lysis via complement.	Primary T-lymphocytes from acute infectious mononucleosis	[60]
	Herpesvirus saimiri (HVS)	Herpesvirus	Enhanced	HVS encodes CD59 mimic protein to evade CDC.	BALB/3T3	[112,113]
	Kaposi’s sarcoma associated herpesvirus (KSHV)	Herpesvirus	Enhanced	Downregulation by KSHV to confer CDC resistance.	Human umbilical vein endothelial cells, microvascular endothelial cells	[114]
	Human Herpesvirus-7 (HHV-7)	Herpesvirus	Enhanced	HHV-7 infection upregulates CD59 to confer partial CDC resistance.	SupT1, PBMC	[115]
	Human T-cell lymphotropic Virus Type 1 (HTLV-1)	Retrovirus	Enhanced	Incorporated into viral particles and confers CDC resistance.	MT-2 cells,	[110]
	Porcine endogenous retrovirus (PERV)	Retrovirus	-	Incorporated into viral particles but is not sufficient for CDC resistance.	ST-IOWA porcine cells	[116]
	HIV-1	Lentivirus	Enhanced	1. Incorporated into viral particle upon the budding; 2. Incorporation confers ADCML and CDC resistance; 3. Decreased CD59 expression upon HIV-1 infection in CD4+ alveolar macrophages; 4. Co-localizes with gp120/gp41 within lipid rafts.	CEM, H9, U937, CHO, Jurkat, alveolar macrophages	[117,118,119,120,121,122,123]
	Hepatitis C Virus (HCV)	Flavivirus	Enhanced	Selective incorporation into viral particles and confers ADCML resistance.	Huh7.5.1 cells	[124,125]
	Dengue Virus (DENV)	Flavivirus	Restricted	Decreases MAC assembly to reduce tissue damage in Dengue Fever (DF)	PBMC	[126,127]
			Enhanced	Monocytes are more susceptible to DENV infection.		
	Respiratory Syncytial Virus (RSV)	Orthopneumovirus	-	Incorporated into virus filaments.	HepG2 cells	[128]
	Influenza A virus (IAV)	Orthomyxovirus	Enhanced	Increases lung inflammation and neutrophil and CD4+T-cell infiltration.	CD59a KO mice,	[129]
	Infectious Bronchitis Virus (IBV)	Coronavirus	Enhanced	Associated with virions and downregulated upon infection to facilitate particle release and resist CDC.	H1299, Vero, DF1 cells	[130]
	Echovirus	Picronavirus	Enhanced	Facilitates infection but not virus binding.	Rhabdomyosarcoma cells	[131]
	Vaccinia Virus (VV)	Poxvirus	Enhanced	Incorporated into viral particle to evade CDC.	RK13, CV-1, HeLa Aortic rat endothelial cells	[132,133]
	Parainfluenza Virus 5 (PIV5)	Paramyxovirus	Enhanced	TGF-b treatment increases CD59 expression in PIV5 progeny virions conferring CDC resistance.	CV-1, MDBK, Vero, A549, HeLa cell Lines	[134]

“Enhanced” denotes viral infection being increased by CD59 or uPAR; “Resistant” denotes viral infection being decreased by CD59 or uPAR. “-” denotes no or minimal effects of CD59 or uPAR on viral infections. CDC—complement dependent cytolysis; ADCML—antibody dependent complement-mediated lysis.

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
