# Peer review of "Relating GPI-Anchored Ly6 Proteins uPAR and CD59 to Viral Infection"

_viruses, 2019, doi:10.3390/v11111060_

Round 1
Reviewer 1 Report
I find that the review written by Yu et colleagues is easy to read and is properly organised. In short, it is a good job. I have no major objections, and only some minor ones:
1. What does "other atlas" mean in Table 1?
2. I think the authors abuse of the acronyms and do not always present them in an adequate way: uPA is defined in line 121 but appears previously in line 119, and CSF, which appears in line 151, does not appear previously defined.
Author Response
Reviewer #1
I find that the review written by Yu et colleagues is easy to read and is properly organised. In short, it is a good job. I have no major objections, and only some minor ones:
We thank the reviewer for his/her positive and helpful comments. See below our response point by point.
1. What does "other atlas" mean in Table 1?
Response: We thank the reviewer’s careful review of our work and have changed the typo “atlas” into “alias”.
2. I think the authors abuse of the acronyms and do not always present them in an adequate way: uPA is defined in line 121 but appears previously in line 119, and CSF, which appears in line 151, does not appear previously defined.
Response: We absolutely agree with the reviewer and have checked all the acronyms used to make sure that they are proper defined when they first appeared.
Reviewer 2 Report
In table 1, it is said that Ly6K is expressed in keratinocytes, it should be testis.
Ly6K protein is highly expressed in testis. This should be correctly referenced.
Author Response
We agree with the reviewer 3 that LY6K is expressed in testis. Given that LY6K is also expressed in keratinocytes, we have included both in Table 1 - see the red highlight in the attached Word file. We have made corresponding changes in references to reflect these changes - see references 7 and 14-16.
Reviewer 3 Report
Manuscript ID: viruses-631812
Type of manuscript: Review
Title: Relating GPI-anchored Ly6/uPAR and CD59 Proteins to Viral Infection
Authors: Jingyou Yu, Vaibhav Murthy, Shan-Lu Liu*
The review presented by Jinyou and colleges revises the role of uPAR and CD59 on viral infections. The paper is well written and provides sufficient information. Some minor suggestions are highlighted below.
Page 3 line 47. The authors mention two conserved motives bearing Cs and mention that their functions “remains incompletely defined”. Please, provide some functional information (as tentative as these might be) or refer to them as conserved C residues not as motives. As it is, the concept of a conserved motif does not apply.
Page 5 figure 1b. The color chosen for the conserved N makes it difficult to see on top of the green protein structure.
Page 6 line 94. It seems that there is an extra space right before “Interestingly,...”
Page 7 line 146. It seems that there is an extra space right after “...HIV patients”
Table 3. It might be due to the peer review version but some cell needs formatting
Table 3. I’m assuming “resistant” means that uPAR or CD59 provide resistance to the viral infection and “Enhanced” that either of them facilitates the infection. Please provide a better explanation in the table legend. Likewise, indicate what “-” means.
Author Response
Reviewer #3
The review presented by Jinyou and colleges revises the role of uPAR and CD59 on viral infections. The paper is well written and provides sufficient information. Some minor suggestions are highlighted below.
We thank the reviewer for his/her positive and helpful comments. See below our response point by point.
Page 3 line 47. The authors mention two conserved motives bearing Cs and mention that their functions “remains incompletely defined”. Please, provide some functional information (as tentative as these might be) or refer to them as conserved C residues not as motives. As it is, the concept of a conserved motif does not apply.
Response: We modified the original sentence by saying that "The “LXCXXC” motif was thought to be the binding site for transition metal ions while the function of the “CCXXXXCN” motif was less well defined".
Page 5 figure 1b. The color chosen for the conserved N makes it difficult to see on top of the green protein structure.
Response: We changed the color into red.
Page 6 line 94. It seems that there is an extra space right before “Interestingly,...”
Response: We deleted the extra space.
Page 7 line 146. It seems that there is an extra space right after “...HIV patients”
Response: We deleted the extra space.
Table 3. It might be due to the peer review version but some cell needs formatting
Response: We checked through and have made changes as appropriate.
Table 3. I’m assuming “resistant” means that uPAR or CD59 provide resistance to the viral infection and “Enhanced” that either of them facilitates the infection. Please provide a better explanation in the table legend. Likewise, indicate what “-” means.
Response: We rephrased/redefined the wording "Enhanced, Resistant and –" as per the reviewer’s suggestions.